

# Modeling compound flood risk and risk reduction using a globally-applicable framework: A case study in the Sofala region

Dirk Eilander[1,2], Anaïs Couasnon[1,2], Frederiek C. Sperna Weiland[2], Willem Ligtvoet[3], Arno Bouwman[3], Hessel C. Winsemius[2], Philip J. Ward[1]

[1]Institute for Environmental Studies (IVM), Vrije Universiteit Amsterdam, Amsterdam, The Netherlands
[2]Deltares, Delft, The Netherlands
[3]PBL Netherlands Environmental Assessment Agency (PBL), The Hague, The Netherlands

*Correspondence to*: Dirk Eilander (dirk.eilander@deltares.nl)

**Abstract.** In low-lying coastal areas floods occur from (combinations of) fluvial, pluvial, and coastal drivers. If these flood drivers are statistically dependent, their joint likelihood might be misrepresented if dependence is not accounted for. However, few studies have examined flood risk and risk reduction measures while accounting for so-called compound flooding. We present a globally-applicable framework for compound flood risk assessments using combined hydrodynamic, impact and statistical modeling and apply it to a case study in the Sofala province of Mozambique. The framework broadly consists of three steps. First, a large stochastic event set is derived from reanalysis data, taking into account co-occurrence of, and dependence between all annual maxima flood drivers. Then, both flood hazard and impact are simulated for different combinations of drivers at non-flood and flood conditions. Finally, the impact of each stochastic event is interpolated from the simulated events to derive a complete flood risk profile. Our case study results show that from all drivers, coastal flooding causes the largest risk in the region despite a more widespread fluvial and pluvial flood hazard. Events with return periods larger than 25 year are more damaging when considering the observed statistical dependence compared to independence, e.g.: 12% for the 100-year return period. However, the total compound flood risk in terms of expected annual damage is only 0.55% larger. This is explained by the fact that for frequent events, which contribute most to the risk, limited physical interaction between flood drivers is simulated. We also assess the effectiveness of three measures in terms of risk reduction. For our case, zoning based on the 2-year return period flood plain is as effective as levees with a 10-year return period protection level, while dry proofing up to 1 m does not reach the same effectiveness. As the framework is based on global datasets and is largely automated, it can easily be repeated for many other regions for first order assessments of compound flood risk.

## 1 Introduction

Floods are associated with the majority and costliest of recorded climate-related hazards over the past 50 years and these disasters disproportionately affect lower-income economies (WMO, 2021). To achieve a substantial reduction in impact of floods it is key to better understand their risk and invest in risk reduction measures (UNDRR, 2015, 2019). Structural measures



such as levees and dams, land use planning, and/or early warning systems in combination with shelters and/or evacuation have proven effective in reducing the impacts of these hazards (UNDRR, 2020; Ward et al., 2017).

Low-lying coastal deltas are especially prone to floods as these areas face flooding from fluvial (discharge), coastal (surge and waves) and pluvial (rainfall) drivers. If these drivers co-occur, they can cause or exacerbate flooding, and are referred to as compound flood events (Wahl et al., 2015; Zscheischler et al., 2020). If statistically dependent, the joint likelihood of these drivers might be misrepresented if dependence is not accounted for (e.g. Ward et al., 2018). Furthermore, physical interactions between these drivers modulate flood levels and are often nonlinear (Bilskie and Hagen, 2018; Serafin et al., 2019). Flood risk assessments in coastal deltas should therefore account for both physical interactions as well as the statistical dependence between flood drivers (Moftakhari et al., 2019). While flood risk assessments for univariate flood drivers are well established and embedded in engineering practices, extending these to multiple flood drivers is a complex undertaking and no generic guidelines exist (Moftakhari et al., 2019; Wu et al., 2021).

Many compound flood studies have either investigated the statistical dependence between drivers or used hydrodynamic models to assess the physical interactions between drivers, while few have combined both aspects to examine extreme flood levels (Serafin et al., 2019; Moftakhari et al., 2019; Gori et al., 2020; Wu et al., 2021). Statistical compound flood studies mostly focus on bivariate driver combinations, for instance surge and discharge (Ward et al., 2018; Couasnon et al., 2020; Hendry et al., 2019), surge and precipitation (Wahl et al., 2015; Bevacqua et al., 2019; Zheng et al., 2013) or surge and waves (Marcos et al., 2019). Few studies have looked at dependence of all four drivers (Nasr et al., 2021; Camus et al., 2021). Hydrodynamic compound flood analyses have mostly been used for a limited number of events at local scales. These studies have focused on interactions between storm surge and discharge (Torres et al., 2015; Olbert et al., 2017; Harrison et al., 2021) or wave setup and discharge (Kupfer et al., 2021), for example to identify where multiple drivers influence water levels, the so-called "transition" zone (Bilskie and Hagen, 2018).

Only a few studies have performed a compound flood risk assessment using combined hydrodynamic, statistical and impact modeling (e.g. Lamb et al., 2010; Bates et al., 2021; Couasnon et al., 2022). Furthermore, compound flood studies that measure the effectiveness of flood risk reduction measures often use simplified flood risk assessments. Torres et al. (2015) performed a feasibility study for a storm surge barrier based on historical scenarios rather than the full risk curve. Lian et al. (2013) assessed the performance of pumps for a large range of return periods based on flood hazard only and did not consider exposure or vulnerability. Van Berchum et al. (2020) assessed multiple flood risk reduction measures based on a full risk assessment, but with a simplified hazard model and under the assumption of statistically independent flood drivers.

The objective of this study is therefore to present a globally-applicable framework for integrated compound flood risk assessments using combined hydrodynamic, impact and statistical modeling and apply it to a case study to evaluate the flood





risk and effectiveness of different risk reduction measures. We achieve this by extending the globally-applicable framework for compound flood hazard modeling from Eilander et al. (2022).

## 2 Methods

The globally-applicable compound flood risk framework is shown in Figure 1, with each of the individual components further discussed in this section as well as a brief introduction to the case study (section 2.1). In order to model compound flood risk,

five main steps are performed: **univariate extreme value analysis** to derive the marginal distributions and boundary conditions for the model event set (section 2.2), multivariate **probabilistic modeling** to derive a stochastic event set (sections 2.3), **flood hazard modeling** using a 2D hydrodynamic model for all model events (section 2.4), **flood impact modeling** using Delft-FIAT for a base scenario and three **risk reduction** measures (section 2.5), and finally calculating **flood risk** based on the interpolated impact from the model event set for each stochastic event (section 2.6).


**Figure 1 Schematic of the globally-applicable compound flood risk framework**

### 2.1 Case Study

We selected the Sofala province of Mozambique as our case study area. The area has recently seen two compound flood events

from tropical cyclones, namely Idai in March 2019 and Eloise in January 2021 which both had a large impact on the area (UN OCHA, 2019, 2021). Furthermore, the proposed flood hazard framework was previously validated for this area for two historical flood events (Eilander et al., 2022). Due to the relative data scarcity, global models have been shown to be useful in supporting decision making in this area (Emerton et al., 2020). The largest city in the Sofala province is Beira, with more than 500,000 inhabitants and a large port connecting the hinterland with the Indian Ocean. While the city itself is mainly threatened

by coastal and pluvial flooding, the deltas of the Pungwe and Buzi rivers are also susceptible to fluvial flooding (Emerton et al., 2020; van Berchum et al., 2020).





## 2.2 Univariate extreme value analysis

To simulate a wide range of plausible (compound) flood events beyond what has been observed in historical time series, we
construct a model event set from combinations of (extreme) flood drivers based on marginal extreme value distributions of
each flood driver. Each event is defined by the following four boundary conditions: discharge at the Pungwe river, discharge
at the Buzi river, rainfall over the model area, and total sea levels, see Figure 2. Note the latter represents the combined wind
setup and storm surge flood drivers, linearly combined with the astronomical tide to obtain total sea levels. For each driver,
we derive one normal (non-extreme) condition and six extreme conditions (2, 5, 10, 50, 100 and 500-year return values). All
combinations of normal and extreme boundary conditions yield a set of 2401 events.

Unless stated differently, marginal extreme values for each driver are based on extreme values distributions fitted to annual
maxima events. Annual maxima are selected from a time series of 42 years based on the hydrological year commencing in
August with a minimal 14 day separation between two events to ensure independent and identically distributed events. The
marginal extreme value distributions are derived by fitting the Gumbel and General Extreme Value (GEV) distributions to the
sampled annual maximum peaks using the L-moments method. The best fit is selected based on the minimum Akaike
Information Criterion (AIC) (Mutua, 1994). For each flood driver, the time series is shown in Figure A1, the fitted distribution
is shown in Figure A2, the return values are listed in Table A1, and the resulting design event time series are shown in Figure
A3. A detailed description of each flood driver, its marginal distribution, and the generation of the model event set is provided
in the following subsections.

### 2.2.1 Discharge

Daily river discharges is simulated with the hydrodynamic CaMa-Flood river routing model version 4.0.1 (Yamazaki et al.,
2011). CaMa-Flood is selected as to our knowledge it is the only global river routing model with an explicit representation of
floodplains, which is important for simulating high discharge events (Zhao et al., 2017). CaMa-Flood uses a 1D river
schematization at a ~10 km resolution to simulate the propagation of discharge based on the local inertial equations (Bates et
al., 2010). The model is forced with runoff data from the ERA5 reanalysis (Hersbach et al., 2020). Time series for the Pungwe
and Buzi rivers are extracted at the boundary of the study region. Other tributaries to the Pungwe at the boundary of the study
region are relatively small and ignored in this study.

For each discharge location, the shape of the fluvial boundary condition is derived by aligning normalized annual maxima
hydrographs with a duration of 14 days centered around the peak and subsequently averaging them. For extreme conditions,
the normalized hydrograph is scaled with the return level as derived from the extreme value distribution. For normal (non-
extreme) conditions, the normalized hydrograph is scaled such that the mean discharge equals that of the mean wet season
(November to April) discharge, see Figure A3.





### 2.2.2 Total sea levels

Total nearshore water levels consist of several components, namely astronomical tide, storm surge and wave setup. The tide and surge components are obtained from the Coastal Dataset for the Evaluation of Climate Impact (CoDEC) (Muis et al., 2020). These components were simulated with the Global Tide and Surge Model (GTSM) version 3.0 (Muis et al., 2020), which is based on the Delft3D Flexible Mesh hydrodynamic model software (Kernkamp et al., 2011). Hourly time series of significant height of wind waves ($H_s$) are extracted at GTSM output locations from the 30 arcmin ERA5 dataset (Hersbach et al., 2020; Bidlot, 2012). The wave setup component is estimated based on $0.2H_s$, which is an often used approximation for (large-scale) studies (US Army Corps of Engineers, 2002; Vousdoukas et al., 2016; Camus et al., 2021). Time series of total water level ($H_{twl}$) are derived by combining the GTSM tide and storm surge components ($H_{st}$) with the wave setup component: $H_{twl} = H_{st} + 0.2H_s$, where $H_s$ is linearly interpolated to 10 min intervals to match the GTSM temporal resolution.

To represent extreme values of tropical cyclone events, the marginal distribution for storm surge is based on a combination of the CoDEC reanalysis data with the COAST-RP dataset (Dullaart et al., 2021). The COAST-RP dataset is based on GTSM storm surge simulations forced with wind and pressure from a synthetic dataset of 3,000 years of tropical cyclone activity (Bloemendaal et al., 2020). The marginal distribution of surge from non-tropical cyclone events is fitted to the annual maxima events from the CoDEC dataset where we filter out tropical cyclones, whereas for surge from tropical cyclone events we use the empirical marginal distribution based on the COAST-RP simulations. The distributions are combined by taking the inverse of the sum of the yearly exceedance frequency of both distributions, similar to Dullaart et al. (2021) but for storm surge instead of combined storm surge and tide levels. The marginal distribution of total sea levels is based on the empirical distribution of extreme total sea level events from the stochastic event set (see section 2.3).

The hydrograph shape for total sea levels is constructed by superimposing a fixed tidal component based on the mean high water spring tide and a normalized non-tidal (surge and wave setup) component, which is scaled such that the total water level peak equals the extreme total sea level. The non-tidal hydrograph component is based on annual maxima peaks from superimposed storm surge and wave setup time series with a duration of 14 days centered around the peak. The peaks are normalized and 'horizontally averaged' such that the hydrograph represents the mean normalized storm magnitude for each duration, see Figure A3.

### 2.2.3 Rainfall

Hourly rainfall times series are derived by spatial averaging ERA5 precipitation reanalysis data over the case study area. We derive extreme values at different durations to construct intensity-duration-frequency (IDF) curves. Annual maxima rainfall intensities are derived for durations of 1, 2, 3, 6, 12 and 24 hours. For each duration the Gumbel extreme value distribution is fitted using the L-moments method. From these IDF curves, design rainfall events with a 24-hour duration were constructed





such that the extreme values at all durations are matched using the alternating block method, see Figure A3. For non-extreme rainfall conditions, the model is forced without rainfall.

### 2.2.4 Relative timing between drivers

To combine the boundary conditions, we determine the relative lag time between the peaks of each variable. We use the discharge at the Buzi river as reference since it is the river with largest extremes in the area and calculated the average lag time between the latter and the other drivers. For this purpose, the 10-minute time series of combined storm surge and hourly wave setup are resampled to daily maxima and the hourly rainfall to daily average rainfall. The relative lag time is found based on the maximum cross correlation for lag times between -10 and +10 days and are shown in Table 1. The rainfall, surge and wave

setup daily maxima tend to occur a few days before high discharges on the Buzi rivers while the discharge peak on the Pungwe tend to occur one day after. We also test the sensitivity of the framework to the observed lag time by comparing the simulated risk with an additional scenario where we assume zero lag time between the peaks of all drivers.

**Table 1. Relative lag time between the Buzi peak discharge and other flood drivers based on maximum cross correlation**

| Flood driver | Relative lag-time to Discharge Buzi peak [days] | Pearson rho [-] |
|---|---|---|
| Discharge Pungwe | +1 | 0.64 |
| Rainfall | -3 | 0.53 |
| Storm surge | -3 | 0.19 |
| Wave setup | -3 | 0.12 |

**2.3 Probabilistic modeling**

Different multivariate statistical approaches have been applied for hydrodynamic flood risk assessments, but typically with only two flood drivers (Moftakhari et al., 2019; Bates et al., 2021; Wu et al., 2021). In this case study, we consider five flood drivers: discharge at the Buzi and Pungwe rivers, rainfall, storm surge and wind setup. We therefore use the approach by Couasnon et al. (2022), in which the marginal distributions of each driver (see section 2.2), the dependence between drivers,

and the co-occurrence of different drivers are simulated separately. The approach consists of three steps. First, we use the annual maxima of each driver to fit a Vine Copula to model their annual joint dependence. Second, we define the rate at which different combinations of drivers co-occur within a given time window. Finally, we sample from the copula model and use the co-occurrence rate to generate the equivalent of 30,000 years of events. For the dependence and co-occurrence analysis, we extend the CoDEC dataset of tide and surge levels with additional simulations to cover the recent extreme events of Idai (2019)





and Eloise (2021) (Eilander et al., 2022). All flood drivers are forced with the same ERA5 meteorological reanalysis, hence providing a coherent dataset for this analysis.

- **Joint dependence of annual maxima.** We use Pair Copula Constructions (PCC), also called Vine Copulas, to model the joint distribution of annual maxima of all drivers because they provide a highly flexible way to model multivariate
dependencies. PCC use the bivariate copula as building blocks to characterize the n-dimensional probability density function and a given structure to define the order in which these building blocks are assembled. More specifically, the n-dimensional copula density is calculated as the product of $n(n-1)/2$ bivariate (conditional) copulas (Bevacqua et al., 2017; Aas et al., 2009). From all the possible mathematically valid decompositions, we select the dimensional vine structure that minimizes the AIC. Each bivariate copula is selected from a set of 10 parametric copula models
from the Elliptical (Gaussian, Student t), Archimedian (Clayton, Gumbel, Frank, Joe) and BB families (BB1, BB6, BB7, BB8) and the independence copula. This ensures that complex behavior, including upper tail dependence, are properly captured and modeled. We fitted the PCC to the time series of annual maxima using the pyvinecopulib package in Python (Nagler and Vatter, 2021).

- **Co-occurring annual maxima.** The rate of co-occurring annual maxima is obtained from the date of observed annual
maxima for all drivers. We assume that annual maxima are co-occurring if they occur within 5 days for discharge drivers and 2 days for rainfall and coastal drivers to account for the different durations of the extreme events. We calculate the number of days between subsequent annual maxima of all drivers and group annual maxima that are co-occurring into a single event. This defines the distribution of the different combinations of co-occurring annual maxima in any given year.

- **Stochastic event set.** To generate the equivalent of 30,000 years of events, we first use the fitted PCC to simulate 30,000 realizations of joint annual maxima. We then combine this with the distribution of co-occurring combinations of annual maxima to create a stochastic event set. In years when all drivers co-occur this leads to a single event, but in most years, we simulate multiple events for which at least one driver is extreme. To derive total water levels, tide, surge, and wave setup are linearly combined. Values of non-extreme drivers are based on a random sample from daily
maxima values below the expected annual return value and a random sample of daily high tide values. The simulated pairs of annual maxima drivers are shown in Figure A4.

## 2.4 Flood hazard modeling

A 2D hydrodynamic SFINCS model is automatically set up with the globally applicable compound flood hazard framework as presented in Eilander et al. (2022). SFINCS is selected as it is designed to efficiently simulate overland flow from compound
flooding at limited computation costs and with good accuracy (Leijnse et al., 2021; Sebastian et al., 2021) and has been validated for two historical events for this case study region (Eilander et al., 2022). Using this setup, we derive a maximum flood depth map for each of the events in the model event set, see section 2.3.



The SFINCS model schematization has three input maps: topography, Manning's roughness, and infiltration; the setup of each
map is shortly described below. The grid is set up at 100 m in the UTM zone 36S projection. Dynamic water level boundary
conditions are set to all coastline cells and discharge boundary points are set at those locations where the Buzi and Pungwe
rivers enter the model domain, see Figure 2.

- The **topography** map is based on MERIT Hydro v1.0 (Yamazaki et al., 2019), which is reprojected using bilinear
  interpolation. As MERIT Hydro elevation data do not represent the bed level of river channels, the river bed levels
  are computed per river segment of ~5 km using a Gradually Varying Flow (GVF) solver based on the common
  assumption that the river should convey a two-year return period discharge without flooding (Neal et al., 2021).
  Besides discharge, the GVF requires a bankfull water surface profile, river width and manning roughness. We first
  create a mask of river cells based on a combination of cells with an upstream area threshold of 25km$^2$ and the 30m
  resolution permanent water mask from the Global River Widths from Landsat (GRWL) dataset (Allen and Pavelsky,
  2018). Riverbank cells are based on all cells adjacent to any river cell. Per segment a low percentile of the height
  above the nearest drain (HAND) of riverbank cells is used to derive the bankfull elevation. This elevation is used to
  approximate the bankfull water surface profile in the GVF. The segment average width is measured as the area of the
  river cells per segment divided by its length. A spatially uniform manning roughness value of 0.03 s$^{1/3}$/m is used. The
  initial river bed level is estimated using Manning's equation and the final bed level is computed by two iterations
  where the river bed level is updated based on the difference between the GVF simulated and observed water surface
  profile similar to Neal et al. (2021). The river depth (relative to the bank full height) is kept constant for the estuarine
  part of the river, which is identified based on a minimum width convergence rate threshold.
- The **Manning roughness** map is based on a spatially uniform value for river cells (0.03 s$^{1/3}$/m) and spatially varying
  values for land cells based on the Copernicus global land cover service dataset (Buchhorn et al., 2020), where the
  same river mask is used as for the topography/bathymetry map. These Manning roughness values are based on Te
  Chow et al. (1988).
- The **infiltration** scheme implemented in SFINCS is based on the Soil Conservation Service Curve Number (SCS-
  CN) method (US SCS, 1965). The method requires a map of potential maximum soil moisture retention to be
  initialized, which is empirically estimated based on soil type, land cover, and antecedent moisture condition. This
  map is based on the 250 m spatial resolution Global Curve Number GCN250 dataset (Jaafar et al., 2019).



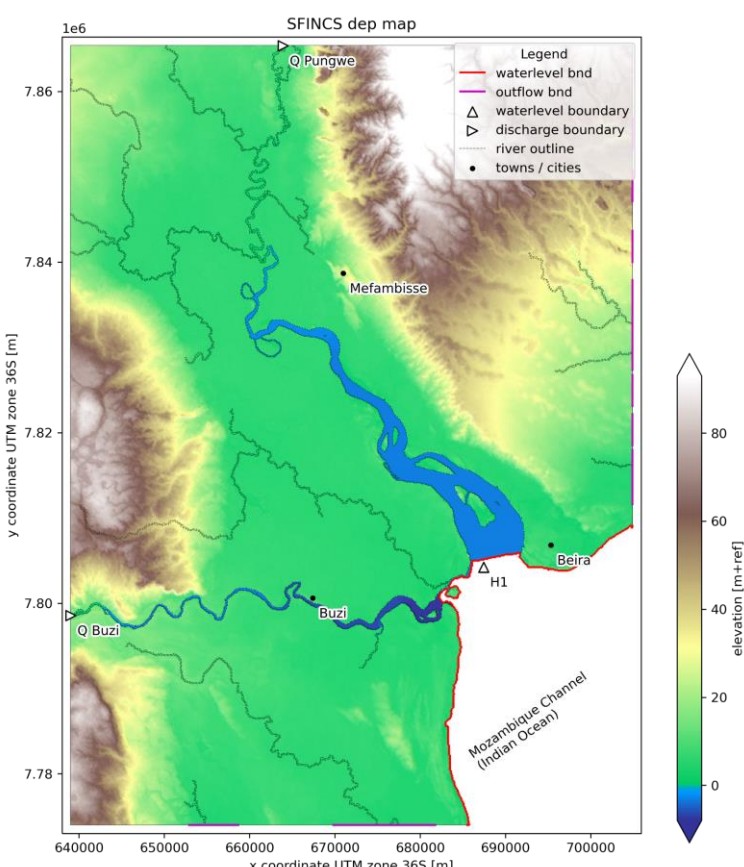

**Figure 2: SFINCS model topography/bathymetry map with the locations of the discharge and water level boundary conditions.**

## 2.5 Flood impact and risk reduction modeling

For each event in the model event set, flood impact is derived using the Delft-FIAT flood impact model (Slager et al., 2016). This model combines the hazard maps with socioeconomic data on exposure and vulnerability to calculate distributed flood impacts per event. Hazard maps are derived as the maximum flood depth from the hydrodynamic simulations. We bias correct all hazard maps based on the limited flood depths simulated for non-flood conditions. This bias in the hazard maps is likely due to inaccuracies in the absolute coastal elevation and river bathymetry. Exposure maps are automatically prepared at the

same resolution as the hazard maps from global data sources using HydroMT (Eilander and Boisgontier, 2022). This procedure and the relevant datasets are described below.

We calculate impact in terms of damage and people affected. The potential damage is estimated per building and based on a country-specific potential damage per person multiplied by the number of residents per building. The country-specific damage

per person is based on residential damage from Huizinga et al. (2017), and additionally accounts for direct non-residential damage (x2.0) and indirect damage (x1.2) using multiplication factors based on various studies (Wagenaar et al., 2019; Koks





et al., 2015). The number of residents per building is obtained by downscaling the gridded population count dataset from WorldPop 2020 UN adjusted data (Bondarenko et al., 2020) based on the Google Open Building building footprints dataset (Sirko et al., 2021). The latter is preprocessed by rasterizing objects with an accuracy larger than 0.7 at a 10 m spatial resolution.

The resulting potential building damage and population counts are shown in Figure 3. The vulnerability is simulated based on a depth-damage function that provides the percentual potential damage as a function of the water depth. Here we use a depth-damage function based on a weighted average of depth-damage functions for different types of buildings from Huizinga et al. (2017). We assume no damage to buildings for water depths smaller than 15 cm, similar to other flood studies (e.g. Wing et al., 2017). The same threshold is used to determine the number of affected people from an event.


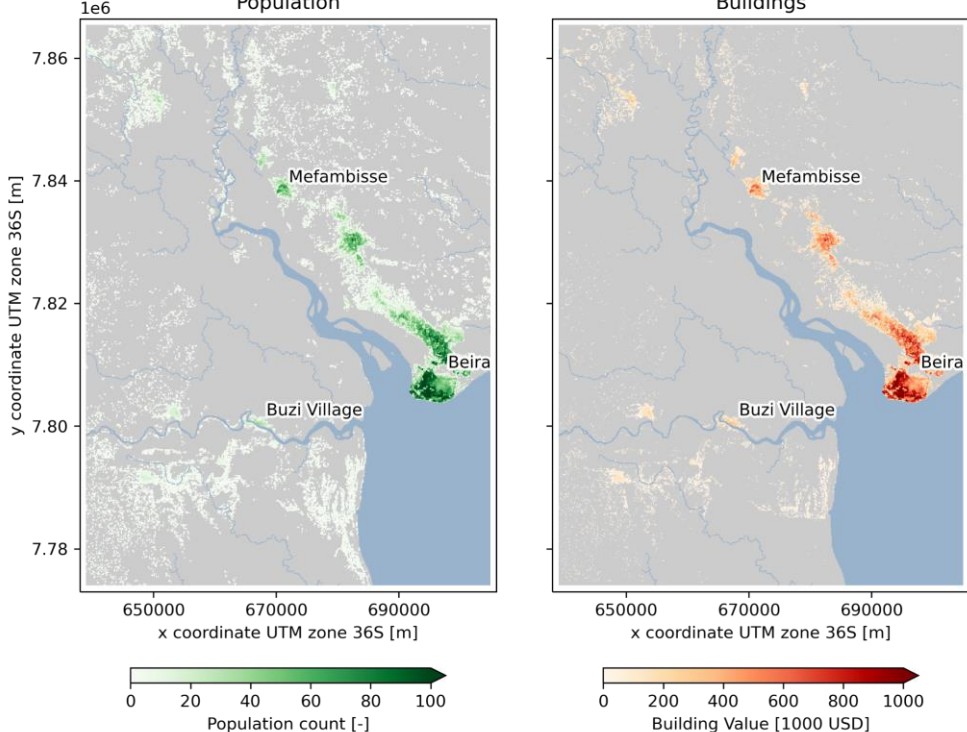

**Figure 3: Estimated population count (left) and building value (right) for the case study area.**

Flood impact is calculated for a base scenario and three scenarios with risk reduction measures: levees, spatial zoning, and dry-proofing of buildings at three different protection levels. All risk reduction measures are implemented in the flood impact

modeling as described below.

- **Levees.** In this scenario we simulate levees with a protection level at a 5, 10 and 50 years return period. No flooding occurs for fluvial or coastal drivers below this level, and above this level we assume complete dike failure. The measure is implemented by correcting the flood levels for scenarios below the protection level. In compound scenarios



with rainfall, a minimum flood depth based on the return level of the univariate scenario with the same rainfall return
level is maintained.

- **Spatial Zoning.** In this scenario exposure (building and inhabitants) within a spatial zone is relocated to an area that is not affected by flooding or made completely flood proof. The spatial zone is defined as the area that is affected (i.e. where the flood depth is larger than 15 cm) in the base scenario at a 2, 5 and 10 years return period. This is implemented by removing all exposure from this area in the impact model.

- **Dry-proofing buildings.** In this scenario flood impact starts at a flood depth larger than the dry proof height of 50, 75 and 100 cm, instead of the 15 cm in the base scenario. This is implemented by setting the percentual damage of the vulnerability (depth-damage) functions to zero for flood depths smaller than the dry proof height.

## 2.6 Flood risk modeling

Flood risk is based on the product of exposure, vulnerability, and hazard over a range of exceedance probabilities. We calculate
the risk in terms of Expected Annual Damage (EAD) and Expected Annual Affected Population (EAAP) as the exceedance probability integral of the flood impact using trapezoidal integration, i.e. the area under the flood impact versus exceedance probability curve (e.g. Ward et al., 2011). The risk is calculated from the empirical exceedance probability for annual damage from the stochastic event set (section 2.4). For each event we derive the flood impact by linear interpolation of the simulated impacts based on its return values.

# 3 Results and discussion

## 3.1 Flood drivers

In this section we present the observed dependence and co-occurrence between all flood drivers. Figure 4 shows the pairwise joint annual maxima, the conditional Kendall's tau correlation coefficient and fitted copula. The joint annual maxima that co-occur with other extremes are highlighted in orange. Each pair is conditioned based on the variables plotted in the panels above
as indicated in the top left of each panel. For six out of the ten pairs of drivers, a significant conditional dependence is found. The strongest dependence is found between the discharge in both rivers and between discharge in the Pungwe river and rainfall ($\tau$=0.43), followed by dependence between surge and wave setup ($\tau$=0.39). Figure 5 shows the distribution of single and compound annual maxima events. In total 141 events are found in 42 years during which at least one driver is extreme. From these events, 45 are compound events with more than 1 extreme, with a maximum duration of 7 days. During three events
(1986, 1992 and 2019) all five drivers co-occurred, one of those being during tropical cyclone Idai in 2019. The number of events increases to 160 (36 compound) if we decrease the maximum time lags between consecutive annual maxima to two days for all drivers, while it decreases to 139 (46 compound) if we increase these time lags to five days for all drivers.

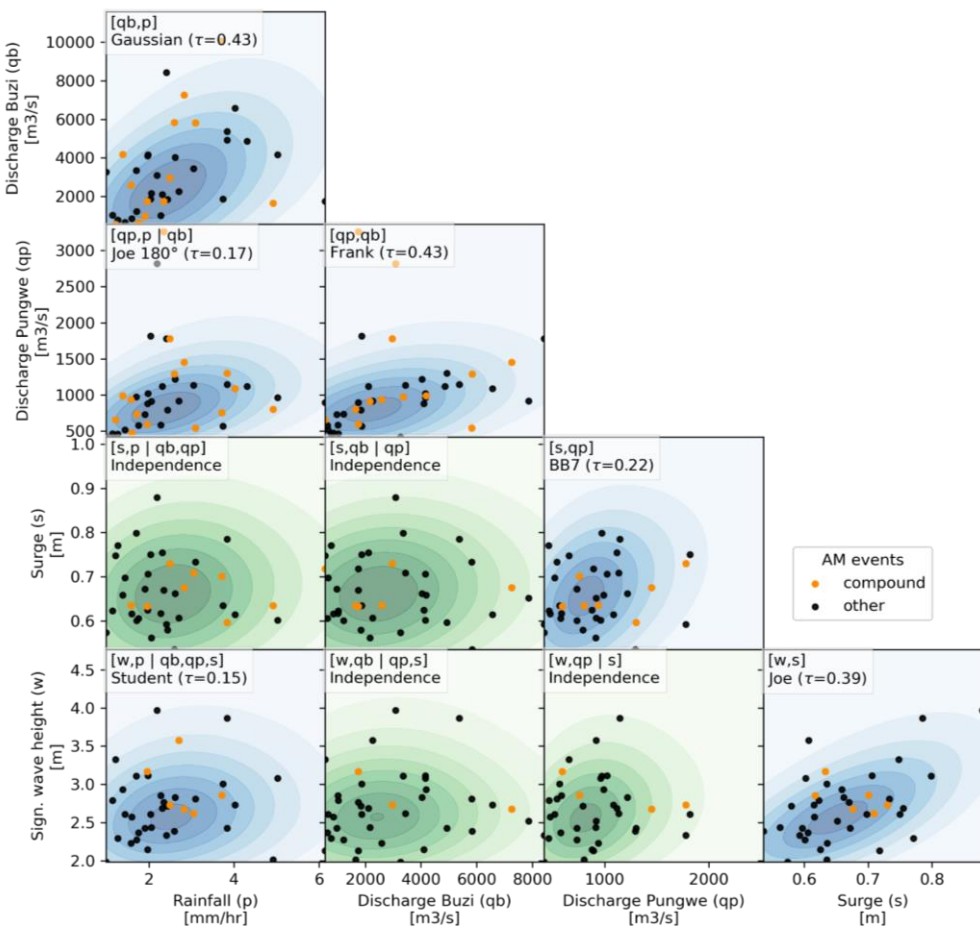

**Figure 4: Conditional dependence between pairs of annual maxima (AM) represented by a Vine copula structure. The dots indicate single (black) or co-occurring (orange) AM events. The background indicates the probability density based on a sample drawn from the vine copula and is colored green for independent and blue for dependent flood driver pairs.**

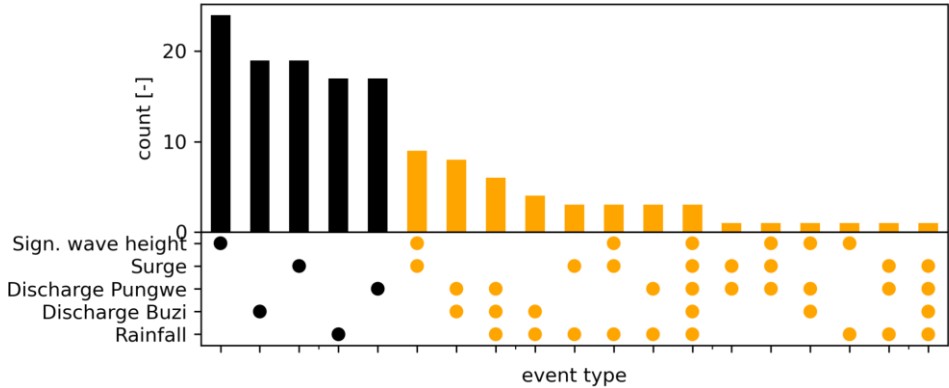

**Figure 5: Distribution of single (black) and compound (orange) events sorted based on occurrence frequency, where the dots indicate**
**the flood drivers combination.**





## 3.2 Flood hazard

In this section we discuss the flood hazard based on the 100-year univariate and compound model events under the assumption of full statistical dependence. Figure 6 shows the pluvial, coastal, and Buzi and Pungwe fluvial flood maps. While the pluvial flooding is most widespread, the flood depths are the smallest among the four univariate hazards. Coastal flooding, on the

other hand, is the most limited in space, but does hit the city of Beira. The fluvial flood maps for both rivers show large spatial extents and large water depths; this is especially the case for the Buzi river flood map where the discharge extremes are the largest. Similar patterns are observed for other return periods. In the left panel of Figure 7, a compound flood hazard map is shown for the event where the 100-year conditions of all drivers co-occur, i.e. the full dependence event. The difference in flood depth between this full dependence compound 100-year flood hazard map and the maximum of each univariate 100-year

flood hazard map shows where physical interactions between the drivers modulate the flood depth, see right panel in Figure 7. In most places the interactions are relatively small compared to the flood depth. In terms of extent, the largest interactions are between the pluvial and fluvial flood drivers. In terms of flood depth, the largest interactions are between the coastal and fluvial drivers. The coastal and fluvial drivers cause the largest increase in flood depths around the upstream end of the Pungwe estuary. Interactions between pluvial and coastal drivers also increase the flood depth with ~20 cm near Beira. Around the

mouth of the Buzi estuary we find that the interactions cause a decrease in flood depth, while further upstream around Buzi town they cause an increase in flood depths. The water levels in the most downstream section of the Buzi river are higher in the compound scenario compared to the 100-year discharge scenario due to backwater effects. However, compared to the 100-year coastal scenario, water levels in the compound scenario are lower, as this river section changes from coastal dominated to discharge dominated. During these high river flow conditions, a lower volume of coastal water enters the river mouth.

Further upstream, the water levels are always discharge dominated and the water levels are larger in compound scenario compared to all single driver scenarios due to backwater effects. This backwater effect causes water levels to increase more and over a larger area  if the peaks of the flood drivers at the boundary happen with zero time lags instead of with the observed time lags, especially in the Buzi river but also in the Pungwe river, see Figure A5.


Figure 6: 100-year flood hazard maps for univariate flood drivers


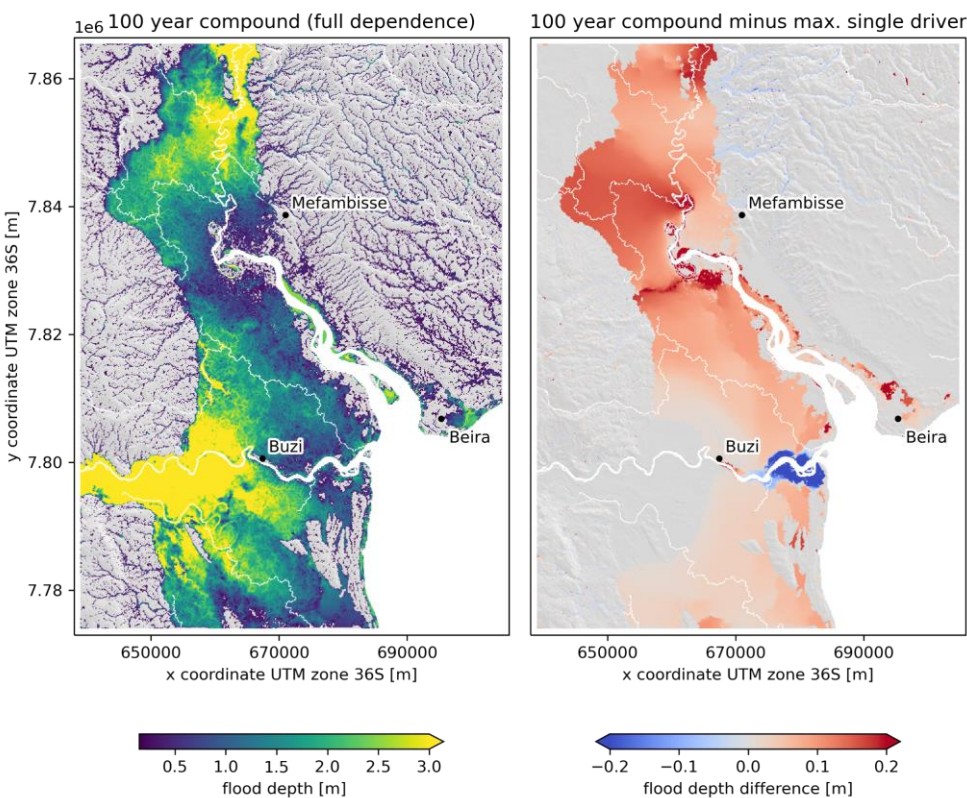

**Figure 7: The 100-year compound flood hazard (assuming full statistical dependence) and the difference between this flood hazard map and the maximum univariate 100-year flood hazard (i.e. maximum from any panels in Figure 6).**

## 3.3 Flood risk

In this section we compare flood risk from univariate flood drivers and compound flood drivers under different assumptions of statistical dependence. The left panel of Figure 8 shows the flood risk profiles, i.e.: the flood impact as a function of the return period, for each univariate flood driver. The univariate risk profiles show that coastal flooding causes the largest risk with an EAD of 40.53 million USD. This is due to the relatively large exposure in coastal areas. The risk curve also shows the steepest incline for events beyond the 100-year return period. This is due to the heavy tail of the marginal distribution for surge related to tropical cyclone activity. Fluvial flooding of the Buzi is more severe in terms of flood depth and extent, but as its floodplains contain less exposure the EAD is lower, at 5.38 million USD. This is similar for fluvial flooding of the Pungwe, where the EAD is 3.06 million USD because of even less exposure. Pluvial flooding does not cause much damage for events up to a 10-year return period, but rapidly increases for more extreme events. This behavior is mostly related to the infiltration capacity of the soil (section 2.4) and flood depth threshold of 15 cm (section 2.5) below which we assume flooding has no impact.





The right panel of Figure 8 shows the compound flood risk profiles under different assumptions of statistical dependence between the joint annual maxima. Each risk profile is based on a stochastic event set with the same number of events based on the observed co-occurrence rates, but with independence, full dependence or observed dependence between the pairs of annual maxima flood drivers. Confidence intervals between the 0.05-0.95 quantiles are derived based on 30 realizations of 1000-year simulations. We report the median risk values and show the confidence interval between brackets. We find a risk based on observed dependence of 58.03 (55.45-60.43) million USD in terms of EAD and 29,990 (28.580-31.230) people in terms of EAAP. This EAD based on observed dependence is smaller than the 58.28 (55.51-61.09) million USD EAD based on full dependence and larger than the 57.71 (56.00-60.01) million USD EAD based on independence. The relative difference in EAD based on independence and observed dependence is 0.55%). While the difference is small and not significant based on the used confidence intervals, the results indicate that taking into account the observed dependence will likely increase flood risk because of an increase in damage from rare events (12% increase at the 100-year return period). In general, the difference in EAD between full dependence and independence is relatively small, namely 0.98%, as the physical interactions between flood drivers mostly occur in locations with little flood exposure. When assuming a zero lag time between flood drivers the risk is 58.19 (55.61-60.59) million USD EAD and 30,080 (28.690-31.330) EEAP. While this assumption results in notable differences in flood hazard (section 3.2), the relative change in risk is small (0.28%) as the differences are at locations with little flood exposure.

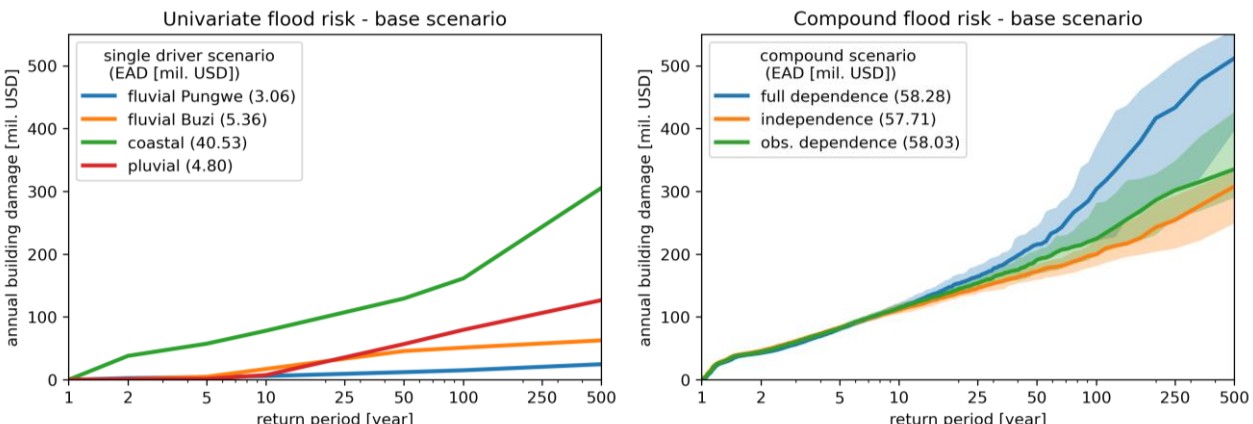

**Figure 8: Flood risk profiles for expected annual damage (EAD) for univariate flooding (left) and compound flooding under different assumptions of statistical dependence (right). The lines show the median and the area around the lines the 0.05-0.95 quantiles based on 30 realizations of 1000-year simulations.**

### 3.4 Flood risk reduction scenarios

Here, we present the effectiveness of three distinct flood risk reduction measures: spatial zoning, dry proofing of buildings, and levees. Figure 9 shows the risk in terms of EAD and EAAP for these measures in absolute values on the left y-axis and as a percentage of the base risk on the right y-axis. Zoning is the most effective risk reduction measure, with a reduction of EAD


by 47.71 million USD (79.0%) and EAAP by ~22,000 (70.4%) people at the middle protection level (i.e. 5-year return period). However, this is also the most drastic as it entails the relocation of 31,800 people living in the 5-year floodplain. In general,

zoning and dry proofing reduce risk across all return periods and act against all flood drivers, levees only reduce risk below the protection level and do not act against pluvial flood drivers. In terms of EAD, the low protection level zoning (2-year return period) and middle protection level levees (10-year return period) measures are similarly effective with a risk reduction of 67.5% and 71.2% respectively, while dry proofing does not reach the same effectiveness across the simulated protection levels. In terms of EAAP, the low protection level zoning (2-year return period) measure, the middle protection level dry proofing

(75 cm) and low protection level levees (5-year return period) measures are similarly effective with a risk reduction of 55.7%, 56.1% and 49.4% respectively.

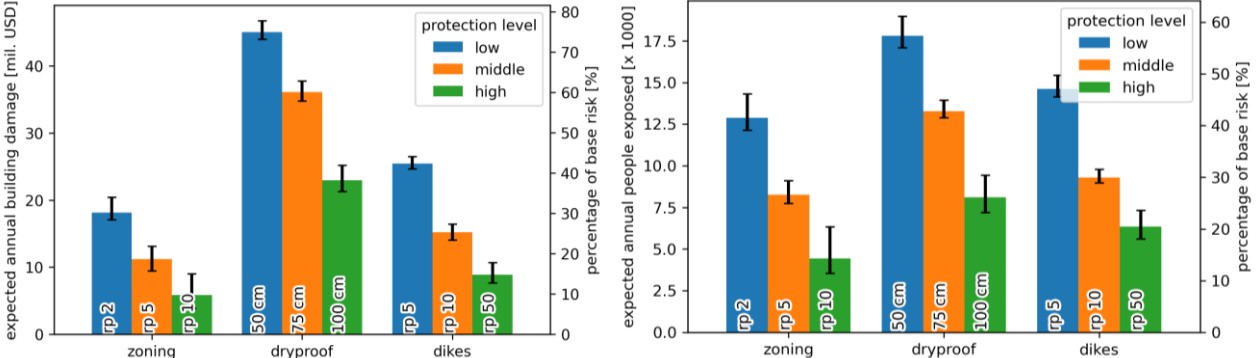

**Figure 9 Compound flood risk for expected annual damage (EAD; left) and expected annual affected population (EAAP; right)**
**under low, middle, and high protection levels of three risk reduction measures: spatial zoning, dry proofing of buildings, and levees.**

### 3.5 Limitations and way forward

While the use of global open source datasets comes with the large benefit of global applicability of the model setup, the accuracy of the input data should be considered. Validation of the model has been limited to a comparison of flood extents as no observed time series of any flood drivers were available (Eilander et al., 2022). The framework allows for integration of

higher quality (local) datasets which, if available, could improve the accuracy of the model. Datasets that would improve the risk assessment are for example a local LiDAR-based DEM, local observations of river bathymetry, observed damages from historical flood events or observed time series of any flood driver.

The change in flood risk when accounting for compound events does not only depend on the dependence between drivers, but

also the co-occurrence rate, duration of and time lags between drivers, and the hydrodynamics of the estuaries (Harrison et al., 2021; Serafin et al., 2019). Here we used the events-based method proposed by Couasnon et al. (2022) to assess flood risk based on the co-occurrence rate of and dependence between annual maxima. Future research should investigate how the selected dependence model and sampling strategy compares to other multivariate dependence models and sampling strategies





(e.g. Zheng et al., 2014; Lucey and Gallien, 2022) to find out which approach is most appropriate for different applications.
The boundary conditions of the hydrodynamic simulations are based on design events with fixed duration and time lags between drivers. Accounting for these in a probabilistic manner would rapidly increase the required number of simulations. Furthermore, we simulated all combinations of flood drivers based on a set of univariate return periods. Alternatively, the selection of simulations could be informed by the multivariate probability density function by selecting only the most likely combination (Moftakhari et al., 2019) or multiple combinations based on weighted random samples (Sadegh et al., 2018) for
each multivariate return period. A brute force approach, which requires fewer assumptions but generally more computational resources (Winter et al., 2020; Wu et al., 2021), could be an interesting alternative to event-based approaches for coastal flood risk assessments with many flood drivers.

Here, we focused on compound flood risk based on current climate conditions. However, to assess risk reduction measures, it
is important to account for changes of environmental, socio-economic and climate conditions. Changes in climate do not only translate to changes in the magnitude of flood drivers, but may also affect the dependence between flood drivers (e.g. Gori et al., 2022). At the same time socio-economic changes will also largely affect flood risk and without action might be the largest driver of change in future flood risk (Winsemius et al., 2015; Neumann et al., 2015).

## 4. Conclusions and recommendations

We applied a globally-applicable compound flood risk framework to the Sofala region of Mozambique, where Beira is located. Using the framework, we compared hazard and risk resulting from different flood drivers; provided an integrated assessment of compound flood risk; and evaluated the risk reduction of three risk reduction measures.

In the base scenario without risk reduction measures and with observed dependence the median EAD is 58.03 (55.45-60.43 at
the 0.05 to 0.95 quantile) million USD and the median EAAP is 29,990 (28.580-31.230) people. Coastal flooding was found to cause the largest risk in the region despite a more widespread fluvial and pluvial flood hazard. The compound flood risk in terms of EAD based on observed statistical dependence was found to be 0.55% larger compared to the assumption of statistical independence, while the assumption of full dependence leads to an overestimate of the flood risk. The small difference is attributed to events with return periods larger than 25 year, which are relatively more damaging, e.g.: 12% at the 100-year
return period. This total difference between full dependence and independence is, however, relatively small due to the limited physical interactions occurring in the simulations between the drivers in areas with significant exposure. Zoning is the most effective risk reduction measure. We find that zoning based on the 2-year return period flood plain is similarly effective to levees with a 10-year return period protection level, while dry proofing up to 1 m does not reach the same effectiveness. For this case we found that the compound flood risk is not sensitive to the time lag between flood drivers. However, this and other
required assumptions in an event-based compound flood risk approach should be further validated in future studies.



As the framework is based on global datasets and is largely automated, it can easily be repeated for many other regions for first order assessments of compound flood risk. We therefore argue that the framework provides a suitable means to improve large scale flood risk estimates.

**Data and code availability**

The scripts and data used to setup the experiments in this study are available from Github at https://github.com/DirkEilander/compound_floodrisk

**Author contributions**

DE, PJW, and AC conceived the idea for this study; DE and AC designed and executed the analysis; DE wrote the manuscript with input from all authors.

**Acknowledgements**

We would like to thank Job Dullaart for providing a dataset with simulated storm surge from stochastic tropical cyclone events. The research leading to these results received funding from the Netherlands Organization for Scientific Research (NWO) in the form of a VIDI grant (Grant No. 016.161.324) and the Future Water Challenges 2 (FWC2) project led by the Netherlands Environmental Assessment Agency (PBL). PJW received funding from the European Union's Horizon 2020 research and innovation programme under grant agreement No 101003276 (MYRIAD-EU).

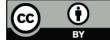



## Appendix A - Supplementary figures

**Figure A1: Time series of the flood drivers considered: discharge at the Buzi and Pungwe rivers, rainfall, daily max storm surge, daily max significant wave heights and total sea levels. Red dots indicate the annual maxima events.**







**Figure A2: Marginal distributions of the flood drivers considered: discharge at the Buzi and Pungwe rivers, rainfall, daily max storm surge, daily max significant wave heights and total sea levels. For surge marginal distributions for non-tropical cyclone (crosses) and tropical cyclone (dots) events are modelled separately and combined.**

455





**Table A1. Extreme values of flood drivers used to setup the hydraulic boundary conditions for the SFINCS model**

| return period [year] | Discharge Buzi [m3/s] | Discharge Pungwe [m3/s] | Rainfall [mm/day] | Wave setup [m] | Surge [m] | Total sea level [m+MSL] |
|---|---|---|---|---|---|---|
| 2 | 2696 | 913 | 2.85 | 0.53 | 0.67 | 4.74 |
| 5 | 5169 | 1406 | 4.43 | 0.64 | 0.75 | 5.05 |
| 10 | 7342 | 1816 | 5.47 | 0.74 | 0.8 | 5.29 |
| 50 | 14286 | 3039 | 7.77 | 1.07 | 1.21 | 5.78 |
| 100 | 18444 | 3726 | 8.74 | 1.28 | 1.74 | 6.02 |
| 500 | 32200 | 5848 | 10.99 | 1.95 | 2.85 | 7.07 |

**Figure A3: Design event time series for non-flood (blue) 2-year flood (orange) and 100-year flood (green) conditions including**
**observed time lag.**




**Figure A4: 10.000 years of simulated (black) and 42 years of observed (red) pairs of annual maxima flood drivers.**

465


**Figure A5: The 100-year compound flood hazard (assuming full statistical dependence) and the difference between this flood hazard map and the maximum univariate 100-year flood hazard assuming zero lag time between the drivers at the model boundary**





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
