# Peer review of "Modeling compound flood risk and risk reduction using a globally-applicable framework: An application in the Sofala region"

_Natural Hazards and Earth System Sciences, 2022_

## Author Comment (AC1)

**Response to reviewer 1**

*General comment*:

The manuscript presents a global framework for assessing flood risk and risk reduction strategies for compound flooding. The framework was applied to Sofala province in Mozambique. The Authors showed that coastal flooding causes the greatest impact regardless of the other drivers.

The manuscript is well-written and with a clear structure. However, there are a few criticalities that need to be addressed.

We would like to thank the reviewer for the thorough review and comments, which we believe have improved the clarity of the manuscript. We are pleased to read that the reviewer considers the manuscript to be well written and clearly structured. Based on the provided feedback to restructure the methods section to better reflect the actual workflow and make it easier to understand how different steps are connected. The intro to section 2 now reads:

*"In order to model compound flood risk, five main steps are performed: univariate extreme value analysis to derive the marginal distributions (section 2.2); flood hazard modeling using a 2D hydrodynamic model for all combinations of derived extreme values (section 2.3); flood impact modeling by combining the simulated flood hazard with exposure and vulnerability data (section 2.4); multivariate probabilistic modeling to derive a large stochastic event set accounting for the joint magnitude and temporal co-occurrence of extremes (sections 2.5); and finally flood risk modeling combining the stochastic event set and simulated flood impacts for a base scenario and three risk reduction scenarios (section 2.6)."*

The novelty of this work compared to the current literature and to previous works from the same Authors is difficult to grasp. In its current form, the manuscript reads as a case study which is not enough. The advice is to highlight how this work addresses the limitations of previous studies and goes beyond what has been already done.

To our knowledge there are only a few studies which combine statistical, hydrodynamic and impact modeling (e.g. Lamb et al., 2010; Bates et al., 2021; Couasnon et al., 2022). However, compared to these studies we provide three major novelties for the advancements of flood risk assessments. First, we go beyond compound risk modeling and include the effectiveness of different adaptation measures. Second, we assess compound flood risk for more than two drivers. Third, our approach is based on data, methods and models that are globally applicable which makes it suitable to scale the approach to other regions. We will amend the intro and discussion to reflect this novelty more clearly. Furthermore, we have changed the

title to emphasize that we present an application of the globally-applicable framework rather than a stand-alone case study.

The global applicability claimed by the Authors is not really proven since they validated and applied it to the same location (line 82). How does this model apply and perform in other locations?

The model framework is indeed only applied for one location, but it has two distinct and critical features which make it globally applicable. Firstly, the schematizations of the hydrodynamic and impact model are automated and based on global datasets only. Secondly, the timeseries of the flood drivers (i.e., the model boundary conditions) are derived from global models. The performance of the model will differ from case to case based on the quality of the input data. We intend to scale the framework to large spatial scales and report on this in future work. In this paper we introduce and apply it for one location. We have added this more explicitly to the discussion in section 3.5.

The description of the probabilistic model needs improvements. First, the Authors should clarify the data used in the vine-copula models, whether they are annual maxima (annual maxima obtained independently in each time series) or whether such data are shifted relative to each other (see table 1). It seems like the annual maxima of 5 different variables occur within a window of +-10 days, which seems a bit unlikely. Moreover, the introduction of a rate of occurrence of annual maxima should be better explained in the relation to the copula model. Why is this necessary? The vine-copula is built to generate sets of dependent variables, including sets in which all of the variables are extremes. This point also relates to the distinction the Authors made between compound events and non-compound events. What does make an event compound? Is this related to the impacts? How can it be defined a priori then?

Our modeling approach does not assume annual maxima (AM) to be co-occurring but instead allows for a flexible description of compound flood drivers, needed to capture the diversity of these events. The probabilistic modeling of the flood drivers considered is based on our analysis of the joint magnitude and temporal co-occurrence of AM of flood drivers. The AM time series are used to (1) fit marginal distributions for each driver; (2) to fit a Vine Copula to simulate the annual joint dependence, and (3) to determine the empirical distribution of temporal co-occurrence based on the dates at which the AM occurred. With the Vine Copula (point 2) we assess the joint dependence in the magnitude of the AM flood drivers. With the empirical co-occurrence distribution (point 3) we assess the chance that the AM flood drivers occur during the same event. To generate the stochastic event set, we first sample from the Vine Copula to obtain a set of correlated AM flood drivers. Then, we sample from the empirical co-occurrence distribution to determine if the sampled AM co-occurred and hence how many events occurred in that year. For example, if the AM

discharge in both rivers and the rainfall co-occur and the surge and wave height AM occur during separate events, we obtain a total of three events (1 joint river and rainfall event, 1 surge event and 1 wave height event). The non-extreme drivers are randomly sampled from all daily values lower than the 1-year return period. We have clarified the approach in 2.5.

For the hydrodynamic modeling, the timing between the boundary conditions is assumed to be constant with respect to the Buzi discharge and is based on maximum cross correlation between daily maxima values of all drivers, as listed in Table 1. Our sensitivity analysis found that a zero lag time assumption changes the simulated water levels (section 3.2) but has a relatively small effect on the simulated flood risk (section 3.3). Therefore, the assumption of a constant timing between flood drivers is acceptable in this case.

We define compound flood events as any combination of flood drivers (extreme and non-extreme) that lead to flooding, in line with the definition used by Zscheischler et al. (2018). Any confusion about this (especially in section 3.1) has been resolved.

*Point-by-point comments:*

Line 36: Do the Authors refer to the joint likelihood or the joint probability? The two concepts are different.

Indeed, these two concepts are different and joint probability should have been used here. This has been modified in line 36 as well as in the abstract.

Line 49: Specify what are the "four drivers".

The four drivers, fluvial (discharge), coastal (surge and waves) and pluvial (rainfall), are mentioned earlier in line 34, but I agree this is not clear and we therefore repeat the drivers in line 49.

Lines 68-74: this paragraph is difficult to read. For example, what is an event set? How is a "model event set from univariate distribution" different from a "stochastic event set from a multivariate probabilistic model"? (see also general comment)

The "model event set" refers to the hydrodynamic and impact model simulations, which have been performed for all combinations of one normal (non-extreme) condition and six extreme univariate conditions (2, 5, 10, 50, 100 and 500-year return values) for all drivers. This step provides a response surface between the magnitude of the flood drivers and the impact obtained for each location of the case study area.

The "stochastic event set" refers to the sample drawn from the Vine Copula model in combination with the co-occurrence distribution. However, we agree with the reviewer that this terminology might be confusing,

and we have rewritten this section to avoid using the term "model event set", see also earlier reply to general comment.

Lines 82: Please further elaborate on the reason why global models are useful in data-scarce regions

We have elaborated on the potential usefulness of global models in the manuscript:

*"In absence of better local data and models, global models have been shown to be useful in supporting risk management in data scarce areas* (Ward et al., 2015)*, for instance for post disaster response in this area by providing bulletins with flood impact forecasts from global models* (Emerton et al., 2020)*. "*

Line 89: Add some information on the boundary conditions. For example, are the boundary conditions generated independently? How are the normal and extreme boundaries selected and combined? Are all normal or are all extremes?

The boundary conditions of the hydrodynamic model simulations are based on all combinations of one normal (non-extreme) condition and six extreme univariate conditions (2, 5, 10, 50, 100 and 500-year return values) for all drivers and generated independently from the joint dependence analysis. We have combined information about the model boundary condition in subsection 2.3.2.

Section Discharge and Total water level. Clarify the link between annual maxima analysis and hydrograph generation. What are the raw data used for annual maxima analysis and how this annual maximum relates to the hydrograph?

The total water level hydrographs are based on annual maxima events of a fixed mean high water spring tide and non-tidal (surge and wave setup) component, which is scaled such that the total water level peak equals the extreme total sea level. The non-tidal hydrograph component is based on annual maxima peaks from superimposed storm surge and wave setup time series. This is clarified in section 2.3.2.

Line 150: why did the Authors use annual maxima if the temporal resolution of the data is hourly?

We calculated annual maxima of rainfall for different durations from 1 to 24 hours, which makes use of the hourly temporal resolution of the data, to construct intensity-duration-frequency curves. This is clarified in section 2.3.2

Line 155: Relative timing between drivers: it seems like the annual maxima of each driver occur around the same time, is it the case? Or the correlation in table 1 is the correlation obtained between the annual max Buzi discharge and the corresponding driver around that period (even if not extreme)? Discuss whether

the timing in Table 1 makes sense. Also, why the Authors selected river discharge at Buzi and not a precipitation event? Precipitation might drive high water in the river unless other processes are of relevance.

An assumption on the relative timing between drivers is required to set up boundary conditions for the hydrodynamic model. Here, we estimate it based on maximum cross correlation of daily maxima values which gives an indication of the relative timing of the co-variability between the drivers. The cross correlation is therefore not only based on events with co-occurring annual maxima as this sample size is simply too small to infer any lag time from. A fixed lag time is a large simplification, but we found based on a sensitivity analysis that it has little impact on the total flood risk in our case. Furthermore, including a varying lag time in the hydrodynamic and probabilistic modeling would drastically increase the computational effort and the assumption of fixed lag time is in line with other compound flood studies (e.g. Bates et al., 2021) .

Discharge is selected as it reflects the rainfall-runoff process in the catchment upstream from the hydrodynamic model domain. Hence there is a lag time between discharge at the model boundary and rainfall within the model domain.

Line 171: Add some discussion on the rate at which different combinations of drivers co-occur. Should not this come from the vine-copula? Please, clarify this second step. (see also general comment)

In our approach, the joint magnitude and temporal co-occurrence of flood drivers are modeled separately, as explained earlier and clarified in section 2.5.

Lines 179: Do you try all the possible vine copulas? It would be good to show somewhere what the vine-copula selected looks like and the associated bi-variate copulas.

The vine copula selected is the one with minimal AIC based on 10 parametric and 1 independence bivariate copula model and selected the one with minimal AIC (section 2.5). We have added a table showing the selected Vine Copula in section 2.5.

Line 189: How is an event defined?

Events are defined based on a maximum lag time between independently sampled annual maxima for each driver. If annual maxima of two drivers occur within the set maximum lag time these are grouped into one event. If the time between two subsequent annual maxima is larger than set maximum lag time, these are modeled as two independent events. Hence, events with single and multiple annual maxima are obtained. We have clarified this in section 2.5.

Line 240: The Authors made a distinction between exposure and vulnerability. However, their definitions are missing. How is exposure defined? How is vulnerability defined? How do they contribute to the impacts? It is not clear which variables have been used to quantify these two concepts.

In this study exposure is expressed by the building value and population. The vulnerability is simulated based on a depth-damage function that provides the percentual potential damage / people affected as a function of the water depth. We have added the following sentence to section 2.4 for clarification:

*"Exposure is here defined by assets and people in the floodplain, the vulnerability as the susceptibility of these assets and people to flooding."*

Line 243: Why is a bias correction needed?

We have clarified this sentence be rephrasing it as:

*"As limited flooding is simulated in the simulation with only non-extreme flood drivers, which does not occur in reality, all hazard maps are bias-corrected with the flood depths of this simulation. This model bias in the hazard maps is likely due to inaccuracies in the absolute coastal elevation and river bathymetry."*

Line 266: Is this return period associated with the univariate case? Is 5-years realistic for the case study? It would be good to justify the choices made.

These return periods are indeed based on the simulations with only coastal or riverine flooding. We have clarified this in the text and added the following sentence to section 2.6 to justify the choice of protection levels:

*"Current flood protection standards are estimated to be around a 2-year return level with the FLOPROS modeling approach* (Scussolini et al., 2016).*"*

From lines 287 – flood drivers. It is a bit unclear how an event is defined and the time series used in the vine-copula models. Why would a compound event be the event in which one variable is extreme? When generating a set of dependent variables, any results in terms of water depths, in this specific case, can be classified as compound (see also general comment)

We agree with the reviewer that any event (both where one or multiple drivers are extreme) is a compound event and this use of the term "compound event" might be confusing. Instead of single versus compound

events we now refer to these as events with single versus multiple co-occurring annual maxima. This sentence in section 3.1 now reads:

*"In total 141 events are found in 42 years during which at least one driver is extreme. From these events, 45 have more than one extreme flood driver and these events have a maximum duration of 7 days."*

Flood Hazard: Please, specify how the 100-year fully dependent event is identified, i.e., how the value of each variable is quantified. Also, are the drivers 4 or 5?

The 100-year fully dependent event is based on the simulation where all marginal 100-year return period events co-occur. There are 5 flood drivers, but four boundary conditions as the surge and wave setup drivers are combined (together with tide which is not considered to be a flood driver) into a single total sea water level boundary condition. We have clarified both in section 3.2:

*"In this section we discuss the flood hazard based on the 100-year univariate and compound event under the assumption of full statistical dependence (i.e. all 100-year flood drivers co-occur). Figure 6 shows the pluvial, coastal (combined surge and waves), and Buzi and Pungwe fluvial flood maps."*

Figure 9: The definition of the percentage of base risk is not fully clear. What is the component of the total risk? Is the total risk different per strategy?

We are not fully sure what the reviewer is referring to with "the component of the total risk", as we do not use the term "total risk" in the manuscript. However, we believe that this refers to the scenario without any adaptation measure which is referred to as the base scenario. The effectiveness of each measure is expressed as the risk reduction relative to the base case, i.e. without any adaptation measure. We have clarified this in section 3.4:

*"Figure 9 shows the risk in terms of EAD and EAAP for these measures in absolute values on the left y-axis and as a percentage of the base risk (i.e., without any risk reduction measure) on the right y-axis."*

References

Bates, P. D., Quinn, N., Sampson, C., Smith, A., Wing, O., Sosa, J., Savage, J., Olcese, G., Neal, J., Schumann, G., Giustarini, L., Coxon, G., Porter, J. R., Amodeo, M. F., Chu, Z., Lewis-Gruss, S., Freeman, N. B., Houser, T., Delgado, M., Hamidi, A., Bolliger, I., McCusker, K., Emanuel, K., Ferreira, C. M., Khalid, A., Haigh, I. D., Couasnon, A., Kopp, R., Hsiang, S., and Krajewski, W. F.: Combined modeling

of US fluvial, pluvial, and coastal flood hazard under current and future climates, Water Resour. Res., 57, e2020WR028673, https://doi.org/10.1029/2020wr028673, 2021.

Couasnon, A., Scussolini, P., Tran, T. V. T., Eilander, D., Muis, S., Wang, H., Keesom, J., Dullaart, J., Xuan, Y., Nguyen, H. Q., Winsemius, H. C., and Ward, P. J.: A flood risk framework capturing the seasonality of and dependence between rainfall and sea levels – an application to Ho Chi Minh City, Vietnam, Water Resour. Res., https://doi.org/10.1029/2021wr030002, 2022.

Emerton, R., Cloke, H., Ficchi, A., Hawker, L., de Wit, S., Speight, L., Prudhomme, C., Rundell, P., West, R., Neal, J., Cuna, J., Harrigan, S., Titley, H., Magnusson, L., Pappenberger, F., Klingaman, N., and Stephens, E.: Emergency flood bulletins for Cyclones Idai and Kenneth: A critical evaluation of the use of global flood forecasts for international humanitarian preparedness and response, International Journal of Disaster Risk Reduction, 50, 101811, https://doi.org/10.1016/j.ijdrr.2020.101811, 2020.

Lamb, R., Keef, C., Tawn, J., Laeger, S., Meadowcroft, I., Surendran, S., Dunning, P., and Batstone, C.: A new method to assess the risk of local and widespread flooding on rivers and coasts, Journal of Flood Risk Management, 3, 323–336, https://doi.org/10.1111/j.1753-318X.2010.01081.x, 2010.

Scussolini, P., Aerts, J. C. J. H., Jongman, B., Bouwer, L. M., Winsemius, H. C., de Moel, H., and Ward, P. J.: FLOPROS: an evolving global database of flood protection standards, Nat. Hazards Earth Syst. Sci., 16, 1049–1061, https://doi.org/10.5194/nhess-16-1049-2016, 2016.

Ward, P. J., Jongman, B., Salamon, P., Simpson, A., Bates, P. D., De Groeve, T., Muis, S., de Perez, E. C., Rudari, R., Trigg, M. A., and Winsemius, H. C.: Usefulness and limitations of global flood risk models, Nat. Clim. Chang., 5, 712–715, https://doi.org/10.1038/nclimate2742, 2015.

Zscheischler, J., Westra, S., van den Hurk, B. J. J. M., Seneviratne, S. I., Ward, P. J., Pitman, A., AghaKouchak, A., Bresch, D. N., Leonard, M., Wahl, T., and Zhang, X.: Future climate risk from compound events, Nat. Clim. Chang., 8, 469–477, https://doi.org/10.1038/s41558-018-0156-3, 2018.

---

## Author Comment (AC2)

**Response to reviewer 2**

This manuscript presents a case study of a modeling framework that could be applied globally to investigate the impact of compound flood risk, at least for initial investigation. The manuscript is for the most part very well written and has a very clear structure.

We would like to thank the reviewer for the review and comments, which we believe will improve the clarity of the manuscript. We are pleased to read that the reviewer considers the manuscript to be mostly well written and clearly structured.

Despite this, there are a small number of comments that need to be addressed. These are listed in page order below but the more important ones are highlighted by *.

Section 2.2.3 Rainfall. Lines 152-153: Design rainfall events with a 24- hour duration were created. Why was this duration chosen? How is this related to catchment response for the chosen area?

The rainfall design events are applied within the SFINCS model domain where the response time of the small tributaries is indeed expected to be in the order of one day. Furthermore, a 24-hour duration is commonly used in flood risk analysis, e.g. in the Soil Conservation Service (SCS) approach (US SCS, 1965). Furthermore, the SCS Curve Number method for infiltration in SFINCS is not applicable to long rainfall events. However, with local rainfall observations (which are not available for this case) an analysis could be done to estimate the typical duration of extreme rainfall events. We have clarified the choice in the methods section and discuss how this boundary condition can be improved in the discussion section.

Line 160: How was the plus/minus ten days determined?

This range is only chosen to calculate the cross-correlation between the drivers. We find a maximum correlation for a lag time of -3 and 1 day. This confirms that the peak in cross correlation is found within the 10-day range. As expected, the cross correlation decreases as expected towards the boundaries of the range, see Figure 1 below. Any range between plus/minus 4 to 10 would result in the same relative lag times and larger time ranges would at some point pick up on cross correlation related to seasonal rather than event-based correlations. We have clarified this in the methods section.

[Figure]

**Figure 1: Cross-correlation between the primary flood driver (i.e., discharge at the Buzi river) and other floods drivers (qp: discharge at the Pungwe; p: rainfall; s: surge; w: total water level).**

Line 176 and throughout the manuscript. The authors use both Pair Copula Constructions and Vine Copula interchangeable throughout the manuscript.

We have edited the manuscript to use Vine Copula consistently in the manuscript.

**\*Section 2** and in particular section 3 (3.2). The authors present the results and talk about inaccuracies (Line 243). However, these are never combined . In Section 3.2 there is a lack of quantifying the statements and relating to the relevant inaccuracies in the data. For example, Line 322-323, the authors states interactions decrease flood depth in the estuary but upstream increases flood depth. By how much and how does this relate to the overall errors in the datasets. This is needed to understand if these changes are significant relative to the data errors. Again Line 326, the authors do not quantify the lower volume of coastal water entering the river mouth and if this is a significant amount.

We appreciate the importance of uncertainty analysis to support any statements made about the physical behavior of the system. In this manuscript we have performed a sensitivity analyses for some of the

assumptions taken in the risk modeling framework, such as the assumption of a constant relative lag time between flood drivers. The sensitivity of the model to uncertainties in the globally applicable model were assessed in a previous paper (Eilander et al. 2022). Based on a comparison with observed flood extents from remote sensing, we found that the model skill is not very sensitive to the river depth, but most sensitive to the Manning roughness and dynamic forcing. We also investigated the sensitivity of the transition zone (i.e. where hydrodynamic interactions between flood drivers increase water levels) to river and estuarine bathymetry based on two events. We found that with a deeper estuary the transition zone in the Pungwe estuary extends further inland, but this change is relatively small compared to the total extent of the transition zone. With sufficient coverage of (new) remote sensing missions such as ICESAT 2 and SWOT, it will become easier to quantify uncertainties in global datasets for local flood studies and go beyond sensitivity analysis. We have added more discussion about the uncertainties in the used datasets and model layers based on relevant literature in section 3.5.

Section 3.3. (Line 345-348). The authors state that the damage caused by pluvial damage is mostly related to the infiltration capacity. Can this be quantified and what are the other factors that influence this.

There appears to be a misunderstanding. We state that the infiltration capacity and the 15cm flood depth threshold for damage results in no damage for events up to 10 years. We have clarified this in the text.

*Section 3.5 Limitations and way forwards (Line 390 - 395). The authors mentioned the accuracy of the input data should be considered. It would be nice to see this point discussed in more detail. This is similar to the point above.

See reply to comment on Section 2 above.

*Line 436-439. This statement sums up the entire manuscript excellently. However, it needs to be stated more strongly throughout the manuscript and include in the manuscript (more clearly) the weaknesses in the approach.

We have included a sentence in the conclusions and abstract to reflect on use and limitations of global datasets and possible ways forward

*"As the framework is based on global datasets and is largely automated, it can easily be repeated for many other regions for first order assessments of compound flood risk. Furthermore, the framework can readily include higher quality (local) datasets to improve the model. We therefore argue that the framework provides a suitable means to improve large scale compound flood risk estimates."*

**References**

US SCS: National engineering handbook, section 4: hydrology, US Soil Conservation Service, USDA, Washington, DC, 1965.

---

## Author Response (AR1)

**Response to reviewer 1**

*General comment*:

The manuscript presents a global framework for assessing flood risk and risk reduction strategies for compound flooding. The framework was applied to Sofala province in Mozambique. The Authors showed that coastal flooding causes the greatest impact regardless of the other drivers.

The manuscript is well-written and with a clear structure. However, there are a few criticalities that need to be addressed.

We would like to thank the reviewer for the thorough review and comments, which we believe have improved the clarity of the manuscript. We are pleased to read that the reviewer considers the manuscript to be well written and clearly structured.

The novelty of this work compared to the current literature and to previous works from the same Authors is difficult to grasp. In its current form, the manuscript reads as a case study which is not enough. The advice is to highlight how this work addresses the limitations of previous studies and goes beyond what has been already done.

We have added several lines to the introduction to highlight the novelties of the paper (line 66). Furthermore, we have changed the title to emphasize that we present an application of the globally-applicable framework rather than a stand-alone case study.

*"Compared to earlier compound flood risk studies, this study provides three advancements. First, it goes beyond compound risk modeling and includes the effectiveness of different adaptation measures. Second, it assesses compound flood risk with a generic approach that is suitable for more than two drivers. Third, the approach is based on global datasets, methods and models, building on the globally-applicable framework for compound flood hazard modeling from Eilander et al. (2023), which makes it globally applicable."*

The global applicability claimed by the Authors is not really proven since they validated and applied it to the same location (line 82). How does this model apply and perform in other locations?

The model framework is indeed only applied for one location, but it has distinct and critical features which make it globally applicable. In this paper we introduce and apply it for one location. We have added discussion about the global applicability in section 3.5 (line 416):

*"In this paper, we applied the framework to one location, but it has two distinct features which make it globally applicable. Firstly, the schematizations of the hydrodynamic and impact model are automated and based on global datasets only. Secondly, the flood drivers (i.e., the model boundary conditions) are derived from global models. These features make it possible to easily apply the framework at a different location. While the use of global open-source datasets and global models comes with the large benefit of global applicability of the model setup, the performance of the model will differ from case to case based on the local quality of the global data and skill of the global models."*

The description of the probabilistic model needs improvements. First, the Authors should clarify the data used in the vine-copula models, whether they are annual maxima (annual maxima obtained independently in each time series) or whether such data are shifted relative to each other (see table 1). It seems like the annual maxima of 5 different variables occur within a window of +-10 days, which seems a bit unlikely. Moreover, the introduction of a rate of occurrence of annual maxima should be better explained in the relation to the copula model. Why is this necessary? The vine-copula is built to generate sets of dependent variables, including sets in which all of the variables are extremes. This point also relates to the distinction the Authors made between compound events and non-compound events. What does make an event compound? Is this related to the impacts? How can it be defined a priori then?

Our modeling approach does not assume annual maxima (AM) to be co-occurring but instead allows for a flexible description of compound flood drivers, needed to capture the diversity of these events. The probabilistic modeling of the flood drivers considered is based on our analysis of the joint magnitude and temporal co-occurrence of AM of flood drivers. The AM time series are used to: (1) fit marginal distributions for each driver; (2) fit a Vine Copula to simulate the annual joint dependence, and (3) determine the empirical distribution of temporal co-occurrence based on the dates at which the AM occurred. With the Vine Copula (point 2) we assess the joint dependence in the magnitude of the AM flood drivers. With the empirical co-occurrence distribution (point 3) we assess the chance that the AM flood drivers occur during the same event. To generate the stochastic event set, we first sample from the Vine Copula to obtain a set of correlated AM flood drivers. Then, we sample from the empirical co-occurrence distribution to determine if the sampled AM co-occurred and hence how many events occurred in that year. For example, if the AM discharge in both rivers and the rainfall co-occur and the surge and wave height AM occur during separate events, we obtain a total of three events (1 joint river and rainfall event, 1 surge event and 1 wave height event). The non-extreme drivers are randomly sampled from all daily values lower than the 1-year return period. We have clarified the approach in section 2.5.

For the hydrodynamic modeling, the timing between the boundary conditions is assumed to be constant with respect to the Buzi discharge and is based on maximum cross correlation between daily maxima values of all drivers, as listed in Table 1. Our sensitivity analysis found that a zero lag time assumption changes the simulated water levels (section 3.2) but has a relatively small effect on the simulated flood risk (section 3.3). Therefore, the assumption of a constant timing between flood drivers is acceptable in this case.

We define compound flood events as any combination of flood drivers (extreme and non-extreme) that lead to flooding, in line with the definition used by Zscheischler et al. (2018). Any confusion about this (especially in section 3.1) has been resolved.

Based on the provided feedback on the methods section we have decided to restructure it to better reflect the actual workflow and make it easier to understand how different steps are connected. The intro to section 2 now reads (line 72):

*"In order to model compound flood risk, five main steps are performed: univariate extreme value analysis to derive the marginal distributions (section 2.2); flood hazard modeling using a 2D hydrodynamic model for all combinations of derived extreme values (section 2.3); flood impact modeling by combining the simulated flood hazard with exposure and vulnerability data (section 2.4); multivariate probabilistic modeling to derive a large stochastic event set accounting for the joint magnitude and temporal co-occurrence of extremes (sections 2.5); and finally flood risk modeling combining the stochastic event set and simulated flood impacts for a base scenario and three risk reduction scenarios (section 2.6)."*

*Point-by-point comments:*

Line 36: Do the Authors refer to the joint likelihood or the joint probability? The two concepts are different.

Indeed, these two concepts are different and joint probability should have been used here. This has been modified in the introduction (line 37) as well as in the abstract (line 10).

Line 49: Specify what are the "four drivers".

The four drivers, discharge (fluvial), surge and waves (coastal) and rainfall (pluvial), are mentioned earlier in line 35, but I agree this is not clear and we therefore repeat these in line 49.

Lines 68-74: this paragraph is difficult to read. For example, what is an event set? How is a "model event set from univariate distribution" different from a "stochastic event set from a multivariate probabilistic model"? (see also general comment)

The "model event set" refers to the hydrodynamic and impact model simulations, which have been performed for all combinations of one normal (non-extreme) condition and six extreme univariate conditions (2, 5, 10, 50, 100 and 500-year return values) for all drivers, i.e., 2401 events. This step provides a response surface between the magnitude of the flood drivers and the impact obtained for each location of the case study area. The "stochastic event set" refers to the sample drawn from the Vine Copula model in combination with the co-occurrence distribution.

However, we agree with the reviewer that this terminology might be confusing, and we have rewritten this section using "simulated flood impacts" rather than "model event set", see also earlier reply to last general comment.

Lines 82: Please further elaborate on the reason why global models are useful in data-scarce regions

We have elaborated on the potential usefulness of global models in the manuscript (line 89):

*"In absence of better local data and models, global models have been shown to be useful in supporting risk management in data scarce areas* (Ward et al., 2015)*, for instance for post disaster response in this area by providing bulletins with flood impact forecasts from global models* (Emerton et al., 2020). *"*

Line 89: Add some information on the boundary conditions. For example, are the boundary conditions generated independently? How are the normal and extreme boundaries selected and combined? Are all normal or are all extremes?

The boundary conditions of the hydrodynamic model simulations are based on all combinations of one normal (non-extreme) condition and six extreme univariate conditions (2, 5, 10, 50, 100 and 500-year return values) for all drivers and generated independently from the joint dependence analysis (see also our penultimate reply). We have combined information about the model boundary condition in subsection 2.3.2.

Section Discharge and Total water level. Clarify the link between annual maxima analysis and hydrograph generation. What are the raw data used for annual maxima analysis and how this annual maximum relates to the hydrograph?

The total water level hydrographs are based on annual maxima events of a fixed mean high water spring tide and non-tidal (surge and wave setup) component, which is scaled such that the total water level peak equals the extreme total sea level. The non-tidal hydrograph component is based on annual maxima peaks from superimposed storm surge and wave setup time series. This is clarified in section 2.3.2.

Line 150: why did the Authors use annual maxima if the temporal resolution of the data is hourly?

We calculated annual maxima of rainfall for different durations from 1 to 24 hours, which makes use of the hourly temporal resolution of the data, to construct intensity-duration-frequency curves. This is clarified in section 2.3.2.

Line 155: Relative timing between drivers: it seems like the annual maxima of each driver occur around the same time, is it the case? Or the correlation in table 1 is the correlation obtained between the annual max Buzi discharge and the corresponding driver around that period (even if not extreme)? Discuss whether the timing in Table 1 makes sense. Also, why the Authors selected river discharge at Buzi and not a precipitation event? Precipitation might drive high water in the river unless other processes are of relevance.

An assumption on the relative timing between drivers is required to set up boundary conditions for the hydrodynamic model. Here, we estimate this lag based on the co-variability between the drivers estimated from the maximum cross correlation of daily maxima. The cross correlation is not only based on events with co-occurring annual maxima as this sample size is simply too small to infer a lag time. A fixed lag time is an additional simplification, but including a varying lag time in the hydrodynamic and probabilistic modeling would drastically increase the computational effort. Furthermore, the assumption of a fixed lag time is in line with other compound flood studies (e.g. Bates et al., 2021). In addition, we performed a sensitivity analysis of this assumption in comparison with a zero lag time assumption and found it has effect on the flood hazard but little impact on the total flood risk in this case (line 388).

*"When assuming a zero lag time between flood drivers the risk is 58.19 (55.61-60.59) million USD EAD and 30,080 (28.690-31.330) EEAP. While this assumption results in notable differences in flood hazard (Section 3.2), the relative change in risk is small (0.28%) as the differences are at locations with little flood exposure."*

Discharge is selected as it reflects the rainfall-runoff process in the catchment upstream from the hydrodynamic model domain. Hence there is a lag time between discharge at the model boundary and rainfall within the model domain.

Line 171: Add some discussion on the rate at which different combinations of drivers co-occur. Should not this come from the vine-copula? Please, clarify this second step. (see also general comment)

In our approach, the joint magnitude and temporal co-occurrence of flood drivers are modeled separately, as explained earlier in our reply to the last general comment and clarified in section 2.5.

Lines 179: Do you try all the possible vine copulas? It would be good to show somewhere what the vine-copula selected looks like and the associated bi-variate copulas.

The vine copula selected is the one with lowest AIC values based on 10 parametric and the independence bivariate copula model (section 2.5). We have added a table showing the selected Vine Copula in section 2.5.

Line 189: How is an event defined?

Events are defined based on a maximum lag time between independently sampled annual maxima for each driver. If annual maxima of two drivers occur within the set maximum lag time these are grouped into one event. If the time between two subsequent annual maxima is larger than set maximum lag time, these are modeled as two independent events. Hence, events with single and multiple annual maxima are obtained. We have clarified this in section 2.5.

Line 240: The Authors made a distinction between exposure and vulnerability. However, their definitions are missing. How is exposure defined? How is vulnerability defined? How do they contribute to the impacts? It is not clear which variables have been used to quantify these two concepts.

In this study exposure is expressed by the building value and population. The vulnerability is simulated based on a depth-damage function that provides the percentual potential damage / people affected as a function of the water depth. We have added the following sentence to section 2.4 for clarification (line 219):

*"Exposure is here defined by assets and people in the floodplain, the vulnerability as the susceptibility of these assets and people to flooding."*

Line 243: Why is a bias correction needed?

We have clarified this sentence be rephrasing it as (line 221):

*"As limited flooding is obtained in the simulation with only non-extreme flood drivers, even though this does not occur in reality, all hazard maps are bias-corrected with the flood depths of this simulation. This model bias in the hazard maps is likely due to inaccuracies in the absolute coastal elevation and river bathymetry."*

Line 266: Is this return period associated with the univariate case? Is 5-years realistic for the case study? It would be good to justify the choices made.

These return periods are indeed based on the simulations with only coastal or riverine flooding. We have clarified this in the text and added the following sentence to section 2.6 (line 298) to justify the choice of protection levels:

*"Current flood protection standards are estimated to be around a 2-year return level with the FLOPROS modeling approach* (Scussolini et al., 2016)*."*

From lines 287 – flood drivers. It is a bit unclear how an event is defined and the time series used in the vine-copula models. Why would a compound event be the event in which one variable is extreme? When generating a set of dependent variables, any results in terms of water depths, in this specific case, can be classified as compound (see also general comment)

We agree with the reviewer that any event (both where one or multiple drivers are extreme) is a compound event and this use of the term "compound event" might be confusing. Instead of single versus compound events we now refer to these as events with single versus multiple co-occurring annual maxima. This sentence in section 3.1 (line 319) now reads:

*"In total 141 events are found in 42 years during which at least one driver is extreme. From these events, 45 have more than one extreme flood driver and these events have a maximum duration of 7 days."*

Flood Hazard: Please, specify how the 100-year fully dependent event is identified, i.e., how the value of each variable is quantified. Also, are the drivers 4 or 5?

The 100-year fully dependent event is based on the simulation where all marginal 100-year return period events co-occur. There are 5 flood drivers, but four boundary conditions as the surge and wave setup drivers are combined (together with tide which is not considered to be a flood driver) into a single total sea water level boundary condition. We have clarified both in section 3.2 (line 334):

*"In this section we discuss the flood hazard based on the 100-year univariate and compound event under the assumption of full statistical dependence (i.e. all 100-year flood drivers co-occur). Figure 6 shows the pluvial, coastal (combined surge and waves), and Buzi and Pungwe fluvial flood maps."*

Figure 9: The definition of the percentage of base risk is not fully clear. What is the component of the total risk? Is the total risk different per strategy?

We are not fully sure what the reviewer is referring to with "the component of the total risk", as we do not use the term "total risk" in the manuscript. However, we believe that this refers to the scenario without any

adaptation measure which is referred to as the base scenario. The effectiveness of each measure is expressed as the risk reduction relative to the base scenario, i.e., without any adaptation measure. We have clarified this in section 3.4 (line 399):

*"Figure 9 shows the risk in terms of EAD and EAAP for these measures in absolute values on the left y-axis and as a percentage of the base risk (i.e., without any risk reduction measure) on the right y-axis."*

**Response to reviewer 2**

This manuscript presents a case study of a modeling framework that could be applied globally to investigate the impact of compound flood risk, at least for initial investigation. The manuscript is for the most part very well written and has a very clear structure.

*We would like to thank the reviewer for the review and comments, which we believe will improve the clarity of the manuscript. We are pleased to read that the reviewer considers the manuscript to be mostly well written and clearly structured.*

Despite this, there are a small number of comments that need to be addressed. These are listed in page order below but the more important ones are highlighted by *.

Section 2.2.3 Rainfall. Lines 152-153: Design rainfall events with a 24- hour duration were created. Why was this duration chosen? How is this related to catchment response for the chosen area?

*The rainfall design events are applied within the SFINCS model domain where the response time of the small tributaries is indeed expected to be in the order of one day. However, provided local rainfall observations (which are not available for this case) an analysis could be done to estimate the typical duration of extreme rainfall events. The limitations of a fixed duration rainfall design events and how this can be improved is discussed in section 3.5. We have clarified the chosen design event duration in the methods section 2.3.2 (line 197):*

*"The duration is based on the approximate response time of the small tributaries based on the Soil Conservation Service (SCS) time to concentration approach (US SCS, 1965)."*

Line 160: How was the plus/minus ten days determined?

*This range is only chosen to calculate the cross-correlation between the drivers. We find a maximum correlation for a lag time of -3 and 1 day. This confirms that the peak in cross correlation is found within the 10-day range. As expected, the cross correlation decreases as expected towards the boundaries of the range, see Figure 1 below. Any range between plus/minus 4 to 10 would result in the same relative lag times and larger time ranges would at some point pick up on cross correlation related to seasonal rather than event-based correlations. We have clarified this in the methods section 2.3.2 (line 204).*

*"This range is only chosen to calculate the cross-correlation between the drivers and decreases as expected towards the boundaries of the range."*

[Figure]

**Figure 1: Cross-correlation between the primary flood driver (i.e., discharge at the Buzi river) and other floods drivers (qp: discharge at the Pungwe; p: rainfall; s: surge; w: total water level).**

Line 176 and throughout the manuscript. The authors use both Pair Copula Constructions and Vine Copula interchangeable throughout the manuscript.

We have edited the manuscript to use Vine Copula consistently in the manuscript.

**\*Section 2** and in particular section 3 (3.2). The authors present the results and talk about inaccuracies (Line 243). However, these are never combined . In Section 3.2 there is a lack of quantifying the statements and relating to the relevant inaccuracies in the data. For example, Line 322-323, the authors states interactions decrease flood depth in the estuary but upstream increases flood depth. By how much and how does this relate to the overall errors in the datasets. This is needed to understand if these changes are significant relative to the data errors. Again Line 326, the authors do not quantify the lower volume of coastal water entering the river mouth and if this is a significant amount.

We appreciate the importance of uncertainty analysis to support any statements made about the physical behavior of the system. In this manuscript we have performed a sensitivity analysis for some of the

assumptions taken in the risk modeling framework, such as the assumption of a constant relative lag time between flood drivers. The sensitivity of the model to uncertainties in the globally applicable model were assessed in a previous paper (Eilander et al., 2023). We have added more discussion about the uncertainties in the used datasets and model layers based on relevant literature in section 3.5:

*"While the use of global open-source datasets and global models comes with the large benefit of global applicability of the model setup, the performance of the model will differ from case to case based on the local quality of the global data and skill of the global models. A validation for two events based on a comparison with flood extents derived from remote sensing and sensitivity analysis of the globally-applicable model has been performed in a previous study (Eilander et al., 2023). Based on a comparison with observed flood extents from remote sensing, we found that the model skill is not very sensitive to the river depth, but most sensitive to the Manning roughness and dynamic forcing. We also investigated the sensitivity of hydrodynamic interactions between flood drivers to river and estuarine bathymetry. Based on that analysis, we found that with a deeper estuary the transition zone (i.e., where hydrodynamic interactions between flood drivers amplify water levels) in the Pungwe estuary extends further inland, but this change is relatively small compared to the total extent of the transition zone.*

*Finally, it should be noted that the framework allows for integration of higher quality (local) datasets which, if available, could improve the accuracy of the model. Datasets that would improve the risk assessment are for example a local LiDAR-based DEM, local observations of river bathymetry, observed damages from historical flood events or observed time series of any flood driver. Furthermore, with sufficient coverage of (new) remote sensing missions, such as ICESAT 2 and SWOT, it will become easier to quantify uncertainties in global datasets for local flood studies and go beyond sensitivity analysis."*

Section 3.3. (Line 345-348). The authors state that the damage caused by pluvial damage is mostly related to the infiltration capacity. Can this be quantified and what are the other factors that influence this.

There appears to be a misunderstanding. We state that the infiltration capacity and the 15cm flood depth threshold for damage results in no damage for low magnitude events up to 10 years. We have clarified this in the text (line 372).

*"The low damage for events up to a 10-year return period is mostly related to the flood depth threshold of 15 cm (Section 2.5), below which we assume flooding has no impact, in combination with the infiltration capacity of the soil (Section 2.4)."*

**\*Section 3.5** Limitations and way forwards (Line 390 - 395). The authors mentioned the accuracy of the input data should be considered. It would be nice to see this point discussed in more detail. This is similar to the point above.

See reply to comment on Section 2 above.

**\*Line 436-439.** This statement sums up the entire manuscript excellently. However, it needs to be stated more strongly throughout the manuscript and include in the manuscript (more clearly) the weaknesses in the approach.

We have included a sentence in the conclusions (line 476) and abstract to reflect on use and limitations of global datasets and possible ways forward

*"As the framework is based on global datasets and is largely automated, it can easily be repeated for other regions for first order assessments of compound flood risk. While the quality of the assessment will depend on the accuracy of the global models and data, it can readily include higher quality (local) datasets where available to further improve the assessment."*

**References**

Bates, P. D., Quinn, N., Sampson, C., Smith, A., Wing, O., Sosa, J., Savage, J., Olcese, G., Neal, J., Schumann, G., Giustarini, L., Coxon, G., Porter, J. R., Amodeo, M. F., Chu, Z., Lewis-Gruss, S., Freeman, N. B., Houser, T., Delgado, M., Hamidi, A., Bolliger, I., McCusker, K., Emanuel, K., Ferreira, C. M., Khalid, A., Haigh, I. D., Couasnon, A., Kopp, R., Hsiang, S., and Krajewski, W. F.: Combined modeling of US fluvial, pluvial, and coastal flood hazard under current and future climates, Water Resour. Res., 57, e2020WR028673, https://doi.org/10.1029/2020wr028673, 2021.

Eilander, D., Couasnon, A., Leijnse, T., Ikeuchi, H., Yamazaki, D., Muis, S., Dullaart, J., Winsemius, H. C., and Ward, P. J.: A globally-applicable framework for compound flood hazard modeling, Nat. Hazards Earth Syst. Sci., 23, 1–40, https://doi.org/10.5194/nhess-23-823-2023, 2023.

Emerton, R., Cloke, H., Ficchi, A., Hawker, L., de Wit, S., Speight, L., Prudhomme, C., Rundell, P., West, R., Neal, J., Cuna, J., Harrigan, S., Titley, H., Magnusson, L., Pappenberger, F., Klingaman, N., and Stephens, E.: Emergency flood bulletins for Cyclones Idai and Kenneth: A critical evaluation of the use of global flood forecasts for international humanitarian preparedness and response, International Journal of Disaster Risk Reduction, 50, 101811, https://doi.org/10.1016/j.ijdrr.2020.101811, 2020.

Scussolini, P., Aerts, J. C. J. H., Jongman, B., Bouwer, L. M., Winsemius, H. C., de Moel, H., and Ward, P. J.: FLOPROS: an evolving global database of flood protection standards, Nat. Hazards Earth Syst. Sci., 16, 1049–1061, https://doi.org/10.5194/nhess-16-1049-2016, 2016.

US SCS: National engineering handbook, section 4: hydrology, US Soil Conservation Service, USDA, Washington, DC, 1965.

Ward, P. J., Jongman, B., Salamon, P., Simpson, A., Bates, P. D., De Groeve, T., Muis, S., de Perez, E. C., Rudari, R., Trigg, M. A., and Winsemius, H. C.: Usefulness and limitations of global flood risk models, Nat. Clim. Chang., 5, 712–715, https://doi.org/10.1038/nclimate2742, 2015.

Zscheischler, J., Westra, S., van den Hurk, B. J. J. M., Seneviratne, S. I., Ward, P. J., Pitman, A., AghaKouchak, A., Bresch, D. N., Leonard, M., Wahl, T., and Zhang, X.: Future climate risk from compound events, Nat. Clim. Chang., 8, 469–477, https://doi.org/10.1038/s41558-018-0156-3, 2018.

---

## Author Response (AR2)

**Review #1**

I want to thank the Authors for taking the time to carefully address the comments made during the first round of revision.

From my side, there is still one point about combining a vine-copula model with a co-occurring model that requires further clarification. The vine-copula approach is used to model the dependence between events so it should already include the concept of co-occurrence. However, here the Authors model the dependence between extremes sampled independently. What then does the correlation between extreme events sampled independently represent? From where does this correlation come? What does the vine-copula model represent in terms of dependence? For example, if the maximum discharge occurs in May and the maximum rainfall in October, what does the correlation between these two variables mean? I suggest that the Authors briefly justify the choice of using the combination of two models (and the meaning of the dependence observed between maximum events) versus modeling the observed dependence between events, e.g., extreme discharge and the associated precipitation event.

Thanks for your feedback and question. We indeed simulate the co-occurrence (empirical distribution) and dependence (Vine Copula) of annual maxima (AM) separately. Because the sample of co-occurring maxima is too small to assess the dependence between drivers, we assume that this dependence can be estimated based on all AM in the same hydrological year. In this case study, apart from the significant wave height AM, the AM of most drivers are within the same season or even month. Therefore, the correlation roughly captures the variability driven by seasonal climatological patterns, see figure A6 below. The benefit of this approach is that it provides information about both aspects of the "compoundness", co-occurrence and dependence, and it is easy to use with more than two drivers. In locations with fewer co-occurring AM or a less distinct wet season the approach might be less applicable.

To capture both the dependence and co-occurrence using a Vine Copula only, we would need to use a different sampling strategy where we sample events conditional to one driver being extreme. For multiple drivers this would require fitting multiple Vine Copulas, each conditioned to a different driver being extreme, and a method to account for co-occurring

extremes that occur in multiple samples. While this approach would also provide information on the magnitude of non-extreme events co-occurring with extremes, this information can currently also not be used in our hydrodynamic simulations because of computational limitations (we simulated all combinations of six extremes and one non-extreme design event for each driver).

We agree with the reviewer that it is important to compare our approach with alternatives, such as the one suggested, to understand under which conditions the assumptions taken are robust (see section 3.5).

We have added the figures to the supplementary information and the following text to section section 3.5 (Line 440):

*"Here, we assume that the dependence can be estimated from all annual maxima. In our case study, where, apart from the significant wave height, the annual maxima of most drivers are within the same season, the correlation roughly captures the variability driven by seasonal climatological patterns, see Figure A6. In locations with fewer co-occurring annual maxima or a less distinct wet season the approach might be less applicable."*

[Figure]

*Figure A6: Day of the year (black dots) and mean day of the year (red line) of the annual maxima of all five drivers. The y-axis indicates the magnitude normalized by the mean annual maxima.*